# Effect of virtual interactions through avatar agents on the joint Simon effect

Yuki Harada[1,2]*, Yoshiko Arima[1,3], Mahiro Okada[3]

**1** Center for Social and Psychological Research of Metaverse, Faculty of Humanities, Kyoto University of Advanced Science, Kyoto, Japan, **2** Developmental Disorders Section, Department of Rehabilitation for Brain Functions, Research Institute of National Rehabilitation Center for Persons with Disabilities, Tokorozawa, Japan, **3** Graduate School of Humanities and Social Sciences, Kyoto University of Advanced Science, Kyoto, Japan

* haradayuuki00@gmail.com

## Abstract

The joint Simon effect refers to inhibitory responses to spatially competing stimuli during a complementary task. This effect has been considered to be influenced by the social factors of a partner: sharing stimulus-action representation. According to this account, virtual interactions through their avatars would produce the joint Simon effect even when the partner did not physically exist in the same space because the avatars are intentional agents. To investigate this, we conducted two joint experiments in which participants controlled avatars as their agents in immersive virtual environments. In Experiment 1, participants were asked to touch a virtual button through their avatars when a target color of an object was presented. The target position was manipulated to be compatible with the button (compatible stimulus) or incompatible with the button (incompatible stimulus). In Experiment 2, the task was identical to that in Experiment 1 except that participants' gaze position was either visualized or not during the task. We hypothesized that visualizing the partner's gaze would enhance the joint Simon effect because gaze is a cue to mentalize others' internal states. The results showed that avatar-based interactions more significantly delayed responses for the incompatible than for the compatible stimulus. However, inconsistent with our hypothesis, visualizing the partner's gaze position did not influence responses for spatially competing stimuli. These results suggest that virtual interactions through avatar agents can produce the joint Simon effect even when the partner does not physically exist in the same space.

## Introduction

To effectively collaborate on joint tasks, it is important to predict the partner's actions, adjust one's own actions, and synchronize both actions. These activities depend on a stimulus-action representation shared with a partner [1]. This shared representation is supported by the joint Simon effect, which refers to an inhibitory response to spatially competing stimuli during a

**Data Availability Statement:** The datasets presented in this study can be found in online repositories. The names of the repository/ repositories and accession number(s) can be found below: 10.17605/OSF.IO/EPWUT.

**Funding:** The author(s) declare financial support was received for the research, authorship, and/or publication of this article. This work was partially supported by the Japan Society for the Promotion of Science KAKENHI (grant numbers 21K02988, 23K12937) for the equipment, payment to participants, and publication fees.

**Competing interests:** The author(s) declare no potential conflicts of interest for the research, authorship, and/or publication of this article.

complementary task. Although the joint Simon effect has been investigated for two decades, its underlying mechanisms remain unclear.

The action co-representation account has explained the joint Simon effect from a social perspective and is composed of three logics. First, spatial incompatibilities between the stimulus and response (e.g., key, button, and mouse click) produce conflicts with the response [2]. For example, responding with a left key to a stimulus presented on the right side was slower than responding with a right key to that [3, 4]. Second, the effect of spatial compatibility occurs at the selection of multiple responses. Specifically, a response that was spatially incompatible with a stimulus was not delayed when there was only one button (go/no-go task [5, 6]). Third, the context of joint tasks triggers the sharing of stimulus-response representations between individuals [1]. In this case, a spatially incompatible response would be delayed, even for the go/no-go task, because each person shares his/her own and his/her partner's stimulus-response representations, that is, two responses are represented in both individuals. Sebanz et al. [7] conducted a complementary task for a pair of participants, where one was asked to push the left key to a specific stimulus, while the other was asked to push the right key to another stimulus (joint Simon task). The results showed delayed responses for keys that were spatially incompatible with the location of the stimulus, even when each participant had only one button.

Consistent with the action co-representation account, the joint Simon effect can be influenced by the perceived intention of a partner. Stenzel et al. [8] conducted a joint Simon task in which participants were asked to respond to a target with a robot partner under two experimental conditions. In one condition, the participants were informed that the robot partner had functions similar to those of a human, such as cameras to detect targets, a neural network to recognize environments, and intelligence to autonomously push the button. In another condition, participants were informed that the robot partner was passively controlled to perform the task. The results showed that only the former condition produced a reliable joint Simon effect. In line with this, the joint Simon effect occurred in situations in which persons believed their partner was a human (not a robot [9]) and in which they performed the joint Simon task with humans (not robots [10]). A recent study found that joint action with a humanoid robot produced the joint Simon effect [11]. This effect was produced even when the robot was controlled by an algorithm, which is partially inconsistent with the results from Stenzel et al. [8]. The incongruency may be because the humanoid used in Ciardo et al. [11] is more human in appearance than that used in Stenzel et al. [8].

The joint Simon effect has also been explained from the perspective of spatial cognitive factors. According to the referential coding account, the presence of a partner can provide a perceiver with spatial cues [12]. Such cues have several advantages such as promoting spatially congruent responses [13] and priming effects for the congruent responses [14], resulting in disadvantages to spatially incongruent responses. Another account has focused on the role of the spatial response coding process [15, 16]. In this account, it is presumed that external stimulus events and internal response events are represented as common formats and that responses are facilitated when the stimulus and response events are similar [17]. These accounts are consistent with previous reports that have shown that the joint Simon effect is produced by spatial visual [18] or vocal information cues [19]. Furthermore, the joint Simon effect has been reported to occur even when social factors are controlled [20].

As economic activities [21, 22] and communications [23] have shifted from physical to remote and virtual environments, recent studies have investigated the effect of interactions in virtual environments on cognitions and behaviors [24–26]. These studies have employed virtual avatars as agents to interact with partners. Recently, Li et al. [27] investigated the effect of virtual interactions using avatars on the joint Simon effect. The results showed that the virtual

interaction produced a statistically reliable joint Simon effect, and that the strength of the effect was weakened when the hands of the avatar were visible, but the body was not. However, to our knowledge, there have been few studies investigating the joint Simon effect in virtual environments. Thus, this study aimed to investigate the effect of the interaction between users' avatars on the joint Simon effect in immersive virtual environments.

This study also focuses on whether the visualization of a partner's gaze can improve the joint Simon effect because the gaze is an important cue to mentalize others' internal states. Since one's eyes can convey mental states such as emotion, interest, and thought [28, 29], the gaze direction captures attention (joint attention: [30]). The joint attention has been observed for not only the gaze of humans but also that of robots or humanoids [31, 32]. Several studies have shown that visualizing a partner's gaze promotes effective interactions in joint task situations [33, 34]. Brennan et al. [35] conducted a joint visual search task in which a pair of participants was asked to collaboratively search for a target from distractors. When the attended location of a partner was visualized, participants adjusted their search to avoid the area being attended by the partner, resulting in a decreased overlap of search patterns with the partner. Gaze visualization has been reported to facilitate social interactions even when the gaze is produced by avatars [36]. Therefore, we hypothesized that the visualizing partner's gaze can enhance the joint Simon effect because the visualization is a cue to read a partner's internal states.

We conducted two experiments in immersive virtual environments to investigate the effect of avatar-based interaction and gaze visualization on the joint Simon effect. In the two experiments, participants always remotely controlled their avatars using three-point tracking (head, left hand, and right hand). In Experiment 1, participants performed go/no-go and joint Simon tasks, where they controlled first-person avatars as their agents. In this case, the joint context would produce the joint Simon effect from the perspective of the co-representation and referential coding accounts. This is because the partner's avatar was intentionally controlled by the partner and can become a cue on spatial coding. In Experiment 2, participants performed the same tasks while their gaze was either visualized or not. Given the co-representation account, the visualization of a partner would enhance the joint Simon effect by promoting mentalizing the partner's internal states. However, the referential coding account predicts that gaze visualization does not affect the joint Simon effect because the gaze has few cues for the spatial position of participants.

## Experiment 1

### Method

**Participants.** A total of 20 undergraduate and graduate students (8 males, 10 females, and 2 others) were recruited from June 14 to October 19, 2023. Sixteen participants were right-handed, 2 were left-handed, and 2 were ambidextrous. Participants' mean age was 19.50 ($SD$ = 1.203) and they had normal or corrected-to-normal visual acuity. Written informed consent was obtained from all participants. This study was approved by the local ethics committee of the Faculty of Humanities of Kyoto University of Advanced Science (approval number: 22H07).

The sample size was determined by previous studies investigating the joint Simon effect [6, 9, 18]. A priori power test was conducted with G*power, in which test family = "F tests," statistical test = "ANOVA: Repeated measures, within factors," effect size ($f$) = 0.31 (Experiment 2 of Dolk et al. [18]), alpha = .05, and power = 0.8 were entered. This test required 24 sample sizes to detect the joint Simon effect. The sample size of Experiment 1 was comparable to the required size.

**Apparatus and materials.** A virtual reality (VR) system was installed in separate rooms (Fig 1A). Each system comprised a head-mounted display (HMD; HTC VIVE Pro Eye), two controllers (HTC VIVE controller 2018), two sensors (HTC SteamVR Base Station 2.0), and a

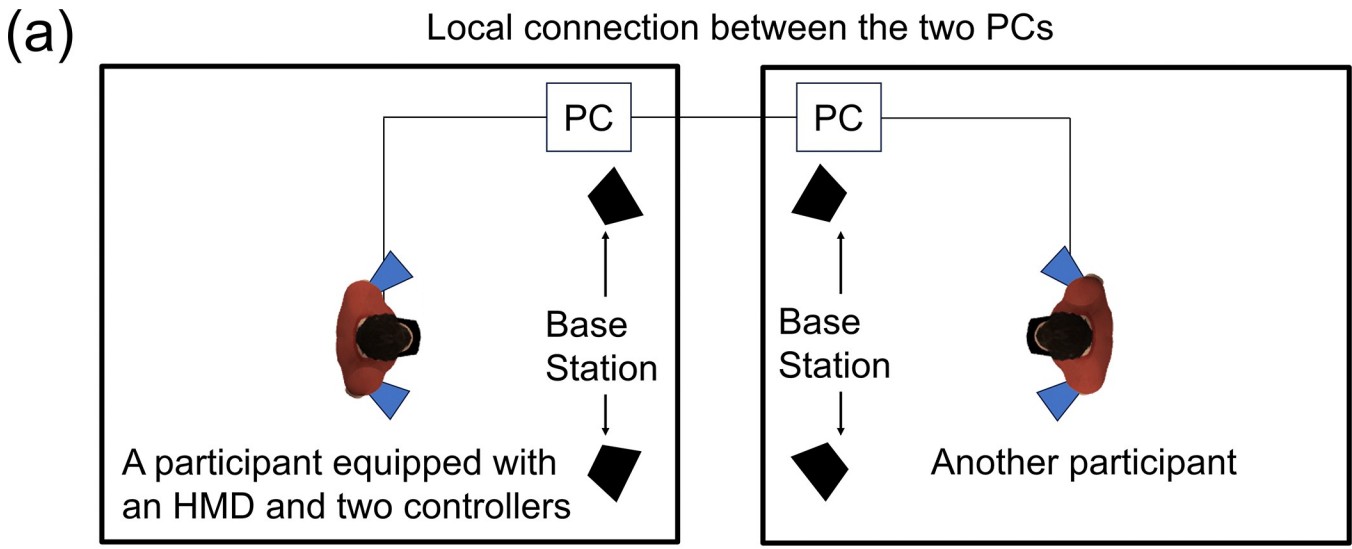

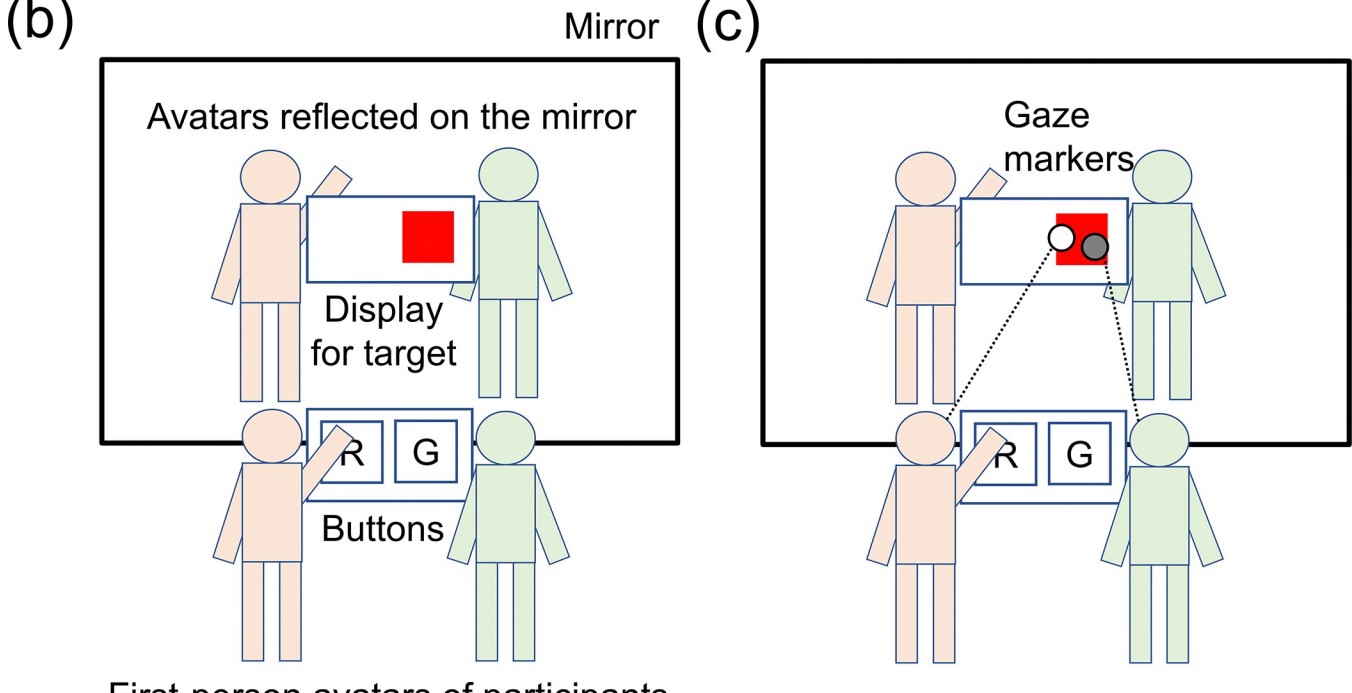

**Fig 1. Illustration of experimental settings and virtual environments.** (a) A pair of VR systems. Each system comprised an HMD, two controllers, two sensors, and a PC. A local network connected the systems. (b) Virtual environments in Experiment 1. (c) Virtual environments in Experiment 2. The environments comprised participants' avatars, a mirror, a display for targets, buttons, and gaze markers (only in Experiment 2).

personal computer. The HMD can measure participants' eye movements during the tasks. The two sensors were placed on the front-left and front-right of participants. The controller was held in each hand. The presentation of virtual stimuli and recoding of responses were conducted using gaming PCs.

The experimental program was created by Unity (2021.3.1f1). A server-client network was constructed between the two VR systems using Unity Netcode for GameObjects. This network allowed participants to interact with each other in immersive virtual environments using their avatars. The avatars had a simple, impersonal appearance, with box-shaped heads, hands, bodies, and feet. The motion of avatars was controlled by an asset "Final IK," which tracked the positions and rotations of three points (an HMD and both controllers) to move those of avatars' heads and both hands. The positions of other body parts were calculated by three-point tracking.

Fig 1B shows the schematic illustration of virtual environments in Experiment 1. The virtual environments comprised participants' avatars, a mirror, and a display for target presentation. Because the position coordinates of the head and both hands corresponded to those of HMD and two controllers, participants could move the avatars as if they were moving their bodies. One avatar existed in the go/no-go task, while two avatars (one is own, the other one is a partner) existed in the joint Simon task. The mirror reflected the avatars of participants, aiming to help participants perceive the status of their avatars. The display for targets was presented between the mirror and the button, where a red or green square was presented on the left or right side. The buttons were located at the front of the participants' avatar and could detect contact with the avatar's hand using the collision enter function in Unity. Participants were asked to push their avatars' hand into the button named "red" when their target was red or that named "green" when their target was green.

**Procedure.**  The experiment was conducted in two quiet rooms and involved two participants who performed the go/no-go and joint Simon tasks in an immersive virtual environment. The participants were positioned on the left or right sides of the environment. The lateral positions were counterbalanced across the participants. After providing informed consent, they were equipped with the HMD and held a controller in each hand.

The experiment was conducted in the order of go/no-go practice, go/no-go test, and joint Simon tasks. Before the practice, participants received a target color (red or green). Participants were asked to push their hand to a corresponding button when the target of the received color was presented, irrespective of target locations, but to ignore it when the non-target color was presented. After the practice (eight trials), the go/no-go task test was conducted. The trial sequence was as follows (Fig 2A). After a three-second count, a black fixation cross was presented at the center of the display for one second. Subsequently, a target square was presented at the left or right side of the display until a response or 2 seconds elapsed. The participants' task was to touch the corresponding button that was virtually set in front of them when the target color was presented. This touch was performed as accurately and quickly as possible, and collisions were detected between the avatar's hand and the button. No feedback was provided for the correctness of responses. After a 1.5-second blank, the next trials began. The target color and location were randomized across trials. The number of total trials was 32: target color (red, green) × target location (left, right) × repetitions (8).

After the go/no-go task, the participants conducted the joint Simon task. The participants were asked to touch a corresponding button when the target color was presented, as in the go/no-go task, but to wait until a partner's avatar touched a corresponding button when a non-target color was presented. The participants knew the target color of a partner via this instruction. A trial sequence was shown in Fig 2B (see S1 Video for the movie of the trial sequence). Similar to the go/no-go task, after the three-second count and fixation cross, a target square

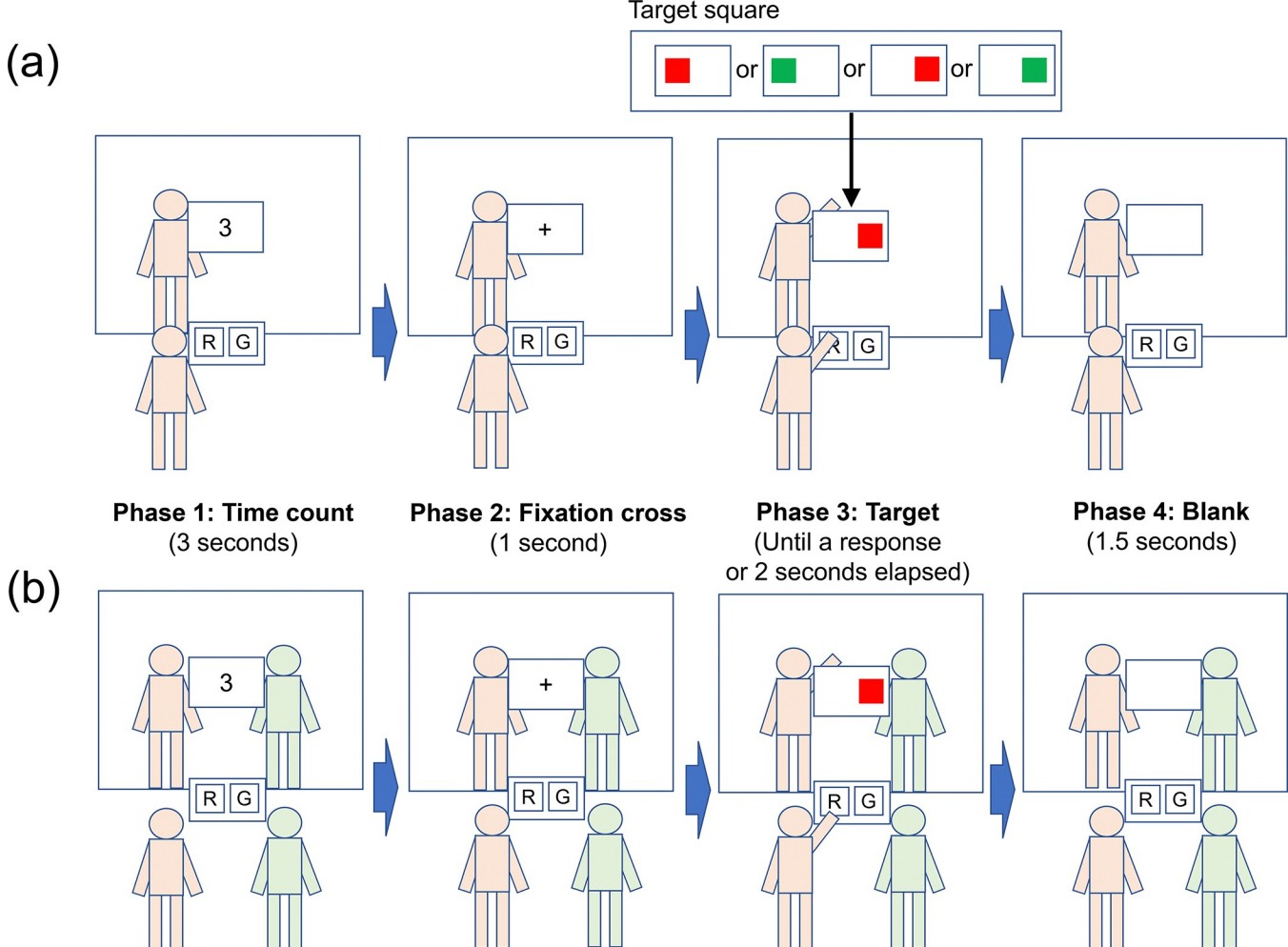

**Fig 2. Trial sequence of the experimental tasks.** (a) The go/no-go task. (b) The joint Simon task. Participants received a target color (red or green) in both tasks. The task was to touch a corresponding button as quickly and accurately as possible when the target of the received color was presented. The illustrations show an example of the incompatible condition.

was presented on the left or right side of the display. The participants were asked to press a corresponding button as quickly and accurately as possible when their target color was presented. Similar to the go/no-go task, after the three second count and fixation cross, a target square was presented at the left or right side of the display. The participants were asked to press a corresponding button as quickly and accurately as possible when their target color was presented. The next trials began after the blank. The target color and location were randomized across trials. The number of total trials was 32: target color (2) × target location (2) × repetitions (8).

The two tasks were the same for the stimulus-response representation but different from the perspective of joint contexts. After the two tasks, the experiment ended.

**Analyses.** The joint Simon effect was evaluated by comparing the response times (RTs) between compatible and incompatible conditions. The compatible and incompatible condition indicates trials in which the target was presented on the same and opposite side of the corresponding button, respectively. We hypothesized that the RTs would be significantly greater in the incompatible condition than in the compatible condition for the joint Simon task but not for the go/no-go task.

Data obtained from one pair of participants were entirely excluded from the analyses due to system troubles that prevented the recording of RTs. RTs obtained from incorrect responses and outlier values were also individually excluded to increase reliability of significance tests following previous studies [37, 38]. The outlier values were defined as RTs more than mean RTs plus two standard deviations [39]. As a result, data obtained from 91.320% of the total trials were analyzed (see S1 Table for the number of valid trials in each condition). One may argue that the outliers should be selected for each condition. However, in the present study, selecting the outlier criteria for each condition would be difficult because the number of repetitions was relatively smaller than those in previous studies [7, 18] (8 repetitions in Experiment 1). In this case, standard deviations become larger, resulting in leaving some outlier RTs. Therefore, we calculated outliers based on the data pooled across the conditions. Moreover, the small repetitions can bias the distribution of RTs. To adjust the distribution, we conducted statistical significance tests on the means of log-transformed RTs as in several previous studies that analyzed RTs [40, 41].

## Results and discussion

Fig 3A shows the mean RTs and standard errors. For the go/no go task, the mean RTs (SDs) were 0.799 (0.208) in the compatible condition and 0.809 (0.209) in the incompatible condition. For the joint task, the mean RTs (SDs) were 0.805 (0.203) in the compatible condition and 0.821 (0.189) in the incompatible condition. To examine whether the joint Simon effect occurred in immersive virtual environments, a two-way repeated measure ANOVA was performed on the means of log-transformed RTs with factors of task (Go/no-go, joint) and compatibility (compatible, incompatible). The main effect of compatibility was significant [$F_{(1, 17)} = 10.263$, $p = .005$, $\eta p^2 = .376$], showing that RTs were significantly longer in the

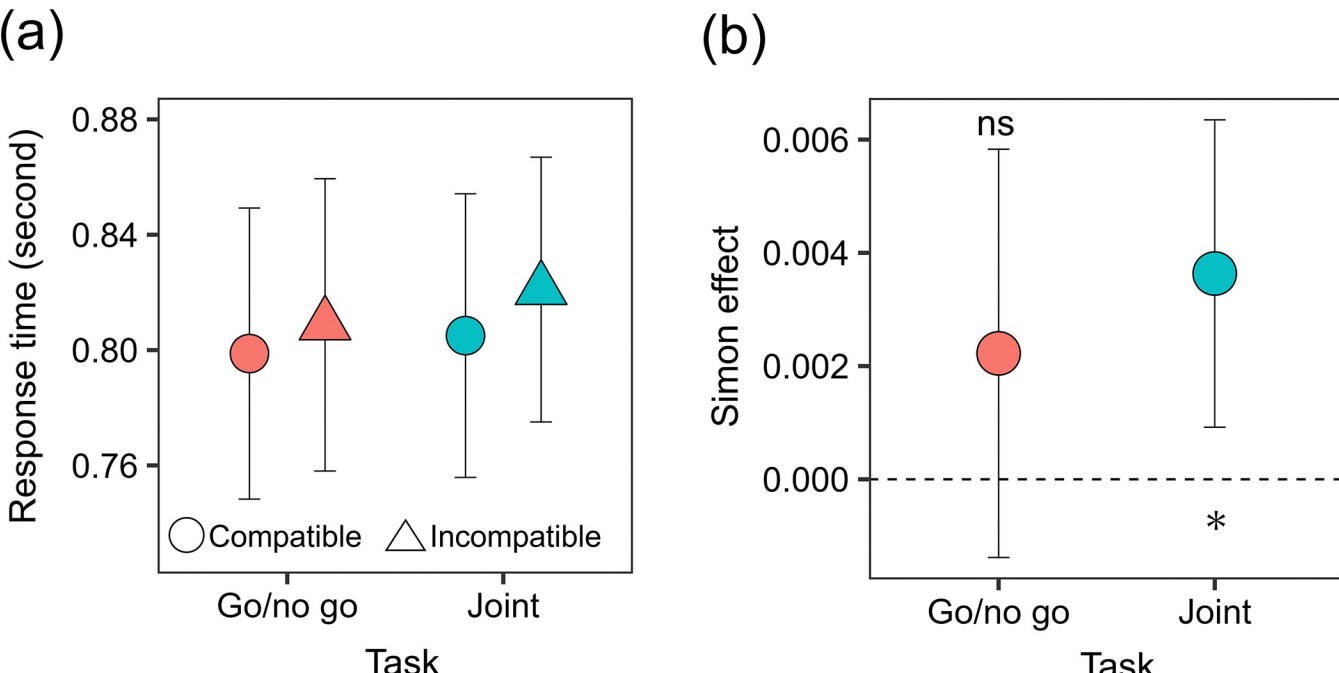

**Fig 3. The effect of spatial compatibility on RTs in Experiment 1.** (a) The mean RTs and the standard errors. (b) The mean Simon effect calculated by (log-transformed RT $_{incompatible}$ −log- transformed RT $_{compatible}$) / log- transformed RT $_{compatible}$. The error bars represent the 95% confidence intervals. The asterisk represents a significant difference between 0 and the value ($p < .05$), while ns represents non-significant difference between them.

incompatible condition than in the compatible condition. Neither the main effect of task nor the two-way interaction was significant [$F(1, 17) = 0.625$, $p = .440$, $\eta p^2 = .035$; $F(1, 17) = 0.323$, $p = .577$, $\eta p^2 = .019$].

Inconsistent with our hypothesis, we obtained a non-significant interaction between the task and compatibility. A possible explanation for these results is that a weaker Simon effect occurred in the go/no-go task. Therefore, we calculated the strength of the Simon effect in each task by (log-transformed RT $_{incompatible}$−log-transformed RT $_{compatible}$) / log-transformed RT $_{compatible}$ (Fig 3B). The mean Simon effects (SDs) were 0.00222 (0.00705) for the go/no go task and 0.00363 (0.00530) for the joint task. A two-tailed paired $t$ test was conducted in each task to examine whether the Simon effect was greater than 0. The significant level was corrected with Holm's method. The results showed that the Simon effect values were significantly greater than 0 in the joint task [$t(17) = 2.825$, $p = .012$] but not in the go/no-go task [$t(17) = 1.301$, $p = .210$]. These results indicate that the avatar-based interaction in the immersive environment produced the joint Simon effect.

## Experiment 2

The effect of gaze visualization on the joint Simon effect was examined through five experimental conditions: go/no-go, joint Simon, partner-gaze, self-gaze, and both-gaze visualization. The go/no-go and joint Simon conditions were identical to those in Experiment 1. In the other three conditions, participants performed the joint Simon task while their partner's and/or their own gaze was visualized. In the partner-gaze condition, the gaze position of the partner was visible. In the own-gaze condition, the own gaze position was visible. In the both-gaze condition, gaze positions of both participants were visible.

### Method

**Participants.**  Forty-four undergraduate and graduate students (23 males and 21 females) were recruited from August 17 to November 30, 2023. Thirty-eight participants were right-handed, 5 were left-handed, and 1 was ambidextrous. Participants' mean age was 20.636 ($SD = 1.350$) and they had normal or corrected-to-normal visual acuity. Written informed consent was obtained from all participants.

The sample size was determined according to previous studies investigating the effect of gaze visualization [32, 34]. A priori power test was conducted with G*power, in which test family = "F tests", statistical test = "ANOVA: Repeated measures, within factors", effect size ($f$) = 0.25, alpha = .05, and power = 0.8 were entered. This test required 34 sample sizes to detect statistically reliable effects. The sample size of Experiment 2 was larger than the required size.

**Apparatus and materials.**  Eye-tracking system equipped on VIVE Pro Eye was used to measure gaze position during the joint Simon task. The streaming data of gaze positions were sent to each VR system using Unity Netcode for GameObjects and utilized to present participants' gaze positions. Fig 1C shows the schematic illustration of virtual environments in Experiment 2. The first-person avatars, mirror, and buttons were identical to those of Experiment 1. The gaze positions were shown as a black or white marker on the mirror in the immersive virtual environments. The gaze markers were presented in the own-gaze, partner-gaze, and both-gaze conditions but not presented in the go/no-go and joint Simon conditions. The gaze marker of own-gaze position was presented in the own-gaze condition, while that of partner-gaze position was presented in the partner-gaze condition. The gaze markers of two participants were presented in both-gaze conditions (their markers could be distinguished due to colors).

**Procedure.** In Experiment 2, participants were instructed to prioritize speed followed by accuracy. This is because requiring quick responses would sharpen the effect of gaze visualization on the RTs. After calibrating the eye-tracking system, participants performed the go/no-go task. Subsequently, they conducted the joint Simon tasks under four conditions manipulated within experimental blocks (see S2 Video for the movie of a trial). Participants were instructed to touch a corresponding button when the target color was presented but to wait until a partner's avatar touched a corresponding button when the non-target color was presented. For the own-, partner-, and both-gaze conditions, it was also explained that their gaze positions, those of their own, that of the partner, or that of both participants, were visualized in the mirror while conducting this task. The trial sequence was the same as that in Experiment 1. The number of total trials was 40: target color (2) × target position (2) × repetitions (10).

The order of four joint conditions was not fully counterbalanced because a reschedule of order was required by the system trouble of gaze visualization. To evaluate the balance of condition order, we calculated the frequency for each of the four joint conditions and order. A chi-square test showed that the order was not significantly unbalanced across the four joint conditions [$\chi^2$ (9) = 1.455, $p$ = .997]. This indicates an almost counterbalanced order.

**Analyses.** A two-way repeated measure ANOVA was performed on the RTs with factors of conditions (go/no-go, joint Simon, partner-gaze, self-gaze, both-gaze conditions) and compatibility (compatible, incompatible). As in Experiment 1, the Simon effect was calculated by (log-transformed RT $_{incompatible}$−log-transformed RT $_{compatible}$) / log-transformed RT $_{compatible}$. A two-tailed paired $t$ test was conducted in each condition to examine whether the Simon effect was greater than 0. The significant level was corrected with Holm's method. To further examine the effect of gaze visualization on the joint Simon effect, a two-way repeated measure ANOVA was conducted on the joint Simon effect with factors of partner-gaze visualization (no, visualized) and own-gaze visualization (no, visualized). RTs obtained from incorrect responses and outlier values were excluded according to the criteria as in Experiment 1. The number of valid trials in each condition is shown in S2 Table.

## Results and discussion

Fig 4A shows the mean RTs and standard errors (see S3 Table for the values of means and *SD*s). A repeated-measures two-way ANOVA was conducted on the log-transformed RTs with factors of five conditions and compatibility. The results showed that the main effect of compatibility was significant [$F$ (1, 43) = 31.860, $p < .001$, $\eta p^2$ = .426], showing that RTs were significantly longer in the incompatible condition than in the compatible condition. Neither the main effect of condition nor the two-way interaction was significant [$F$ (4, 172) = 0.798, $p$ = .528, $\eta p^2$ = .018; $F$ (4, 172) = 1.453, $p$ = .219, $\eta p^2$ = .033].

As in Experiment 1, we calculated the Simon effect (Fig 4B). The mean Simon effects (SDs) were 0.00138 (0.00899) for the go/no go, 0.00412 (0.00729) for the joint, 0.00478 (0.00696) for the partner gaze, 0.00369 (0.00719) for the own gaze, and 0.00433 (0.00887) for the both-gaze conditions. The results of the two-tailed paired $t$ test showed that the Simon effect values were significantly greater than 0 in the joint, partner-gaze, own-gaze, and both-gaze conditions [$t$ (43) = 3.705, $p < .001$; $t$ (43) = 4.507, $p < .001$; $t$ (43) = 3.363, $p$ = .0016; $t$ (43) = 3.200, $p$ = .0026] but not in the go/no-go condition [$t$ (43) = 1.004, $p$ = .321]. These results generally replicated those of Experiment 1.

Experiment 2 manipulated the visualizations of own and partner's gaze individually. To compare the effects of own and partner's gaze visualization, we also conducted a two-way repeated measures ANOVA on the joint Simon effect with factors of own and partner's gaze (absent and present in each factor). The result showed that neither any main effect nor

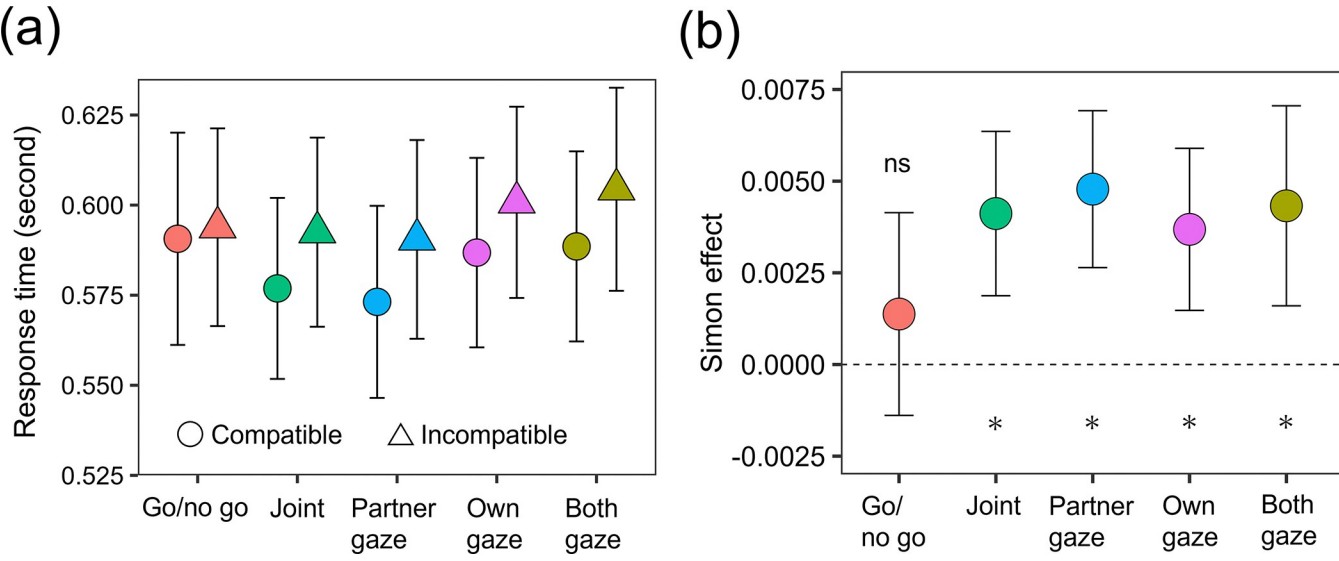

**Fig 4. The effects of spatial compatibility and gaze visualization on RTs.** (a) The mean RTs and the standard errors. (b) The mean Simon effect calculated by (log-transformed RT $_{incompatible}$−log-transformed RT $_{compatible}$) / log-transformed RT $_{compatible}$. The error bars represent the 95% confidence intervals. The asterisk represents a significant difference between 0 and the value ($p < .05$). The ns represents non-significant difference between them.

interaction was significant ($Fs < 0.305$, $ps > .583$, $\eta p^2 s < .007$). Inconsistent with our hypotheses, this suggests that visualizing a partner's or one's own gaze does not influence the joint Simon effect.

## General discussion

This study conducted two experiments in immersive virtual environments to investigate the effects of avatar-based interaction and gaze visualization on the joint Simon effect. Our primary finding is that avatar-based interactions produced the joint Simon effect. Because this effect was observed in both Experiments 1 and 2, it is considered to be robust. The results are consistent with both the action co-representation and referential coding accounts. The action co-representation account assumes that joint tasks have persons share the stimulus-response representations with each other [1], resulting in conflicts to responses for spatially incompatible stimuli. Furthermore, the referential coding account asserts that the presence of a partner offers a spatial cue that guides persons to respond to the button congruent with the cue [12].

Inconsistent with our hypothesis, visualizing a partner's gaze did not influence the joint Simon effect. The results may be explained by two possibilities. One possibility is that visualizing a partner's gaze position may not promote mentalization of the partner's internal states. In daily situations, persons detect another person's eyes [29, 30], recognize the direction of their eyes, and attend to an object that the other person is looking at [42]. In contrast, in Experiment 2, where the partner's gaze position was visualized as a marker on a mirror, which is less common in daily situations. In particular, the detection of the eyes was not required due to visualization methods, which may reduce mentalization of the partner's internal states. Nonetheless, previous studies have shown that visualizing a partner's gaze improves joint task performance, especially for avoiding the duplication of efforts between two persons [33–35]. This duplication avoidance requires one's own behaviors to be adjusted by mind reading. Therefore, manipulating gaze visualization would promote mentalization of the partner's internal states.

Another possibility is that mentalizing a partner' internal states does not necessarily influence the joint Simon effect. Although there is evidence of the effect of social factor on the joint

Simon effect [8, 9, 43], some studies have shown that social factors are insufficient to produce reliable joint Simon effects. For example, cooperating with a human partner in joint tasks did not produce the joint Simon effect when the partner was invisible [44] or when the partner was not present on the left or right side of the participants [6]. These reports highlight that the spatial cues are important to produce the joint Simon effect rather than the social factors. This viewpoint is consistent with the results of Experiment 2.The mean RTs were approximately 200 ms faster in Experiment 2 than in Experiment 1. This would stem from the difference in instructions between the two experiments. The participants were asked to give priority to accuracy followed by speed in Experiment 1 but to quickness followed by accuracy. The reason for the instruction of Experiment 1 was that correct responses were required for analyzing correct RTs. Similarly, the amplitudes of the joint Simon effect were larger in Experiment 2 than in Experiment 1 (Figs 3B vs. 4B), which may stem from the difference in instructions. This would be because fast responses are less likely to contain noise and artifacts.

A potential limitation of Experiment 2 is that it did not eliminate perceived intention of a partner even in the non-gaze condition. Previous studies have achieved this in the control condition by convincing participants that the partner was a non-human [8–10]). However, the Experiment 2 did not accomplish this because the avatar of a partner was present in the four conditions. The avatar presence would convey intention, which may lead to difficulty detecting the effect of gaze visualization. Therefore, further studies are needed to manipulate the avatar presence and gaze visualization individually to investigate this possibility. Another limitation is that we did not obtain the two-way interaction between the compatibility and task because the Simon effect was slightly produced for the go/no go task. Instead, we analyzed whether the strength of the Simon effect (Figs 3B and 4B) was larger than 0. Therefore, the present results show the Simon effect for the joint conditions but do not necessarily show the difference between the go/no go and joint conditions.

It may be important to investigate the effect of gaze visualization on the joint Simon effect from the perspective of social cognitive processing. Gazes convey social cognitive information such as the intention, goal of behaviors, and emotional states [29] by directing gazes to the subject of interest. In joint contexts, gazes can frequently be directed to a partner, which would be a cue to communicate information about joint tasks between persons. Given this, social communications through gaze directions may promote stimulus-response representation. Unfortunately, the present study cannot answer the question because it did not manipulate the direction of gazes and just presented gaze positions as dots. This possibility should be investigated in future studies.

To summarize, our findings revealed that (a) virtual interactions through avatar agents had an effect on the joint Simon effect in an immersive virtual environment, whereas (b) the partner's gaze visualization did not. These results suggest that virtual interactions promote sharing the stimulus-action representation even when the partner does not physically exist.

## Supporting information

**S1 Table. The numbers of valid trials in each condition and participant in Experiment 1.** (XLSX)

**S2 Table. The numbers of valid trials in each condition and participant in Experiment 2.** (XLSX)

**S3 Table. The means and SDs for each condition in Experiment 2.** (XLSX)

**S1 Video. The video of Experiment 1.** This depicts a trial sequence of Experiment 1.
(MP4)

**S2 Video. The video of Experiment 2.** This depicts a trial sequence of Experiment 2.
(MP4)

## Acknowledgments

We thank Kouhei Sano, Riko Nishitani, and Naoya Sumida for their data collections and help-ful discussions.

## Author Contributions

**Conceptualization:** Yuki Harada, Yoshiko Arima, Mahiro Okada.

**Data curation:** Yuki Harada, Yoshiko Arima, Mahiro Okada.

**Formal analysis:** Yuki Harada.

**Funding acquisition:** Yuki Harada, Yoshiko Arima.

**Investigation:** Yuki Harada, Mahiro Okada.

**Methodology:** Yuki Harada, Yoshiko Arima, Mahiro Okada.

**Project administration:** Yuki Harada.

**Resources:** Yuki Harada, Yoshiko Arima.

**Software:** Yuki Harada.

**Supervision:** Yoshiko Arima.

**Visualization:** Yuki Harada.

**Writing – original draft:** Yuki Harada.

**Writing – review & editing:** Yuki Harada, Yoshiko Arima, Mahiro Okada.

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
