## [Decision Letter · Decision Letter 0]

14 Aug 2024

PONE-D-24-19088Effect of Virtual Interactions Through Avatar Agents on the Joint Simon EffectPLOS ONE

Dear Dr. Harada,

Thank you for submitting your manuscript to PLOS ONE. After careful consideration, we feel that it has merit but does not fully meet PLOS ONE’s publication criteria as it currently stands. Therefore, we invite you to submit a revised version of the manuscript that addresses the points raised during the review process. I now have received two expert reviews of your manuscript and although Reviewers concur that the topic of your study is interesting they both highlight major issues that prevent publication in its current form. I myself share most of their concerns. Specifically, both Reviewers point out that the theoretical framework should be broadened (they both provide useful suggestions to this end). In addition, both Reviewers suggest to discuss unexpected results in a revised version and to seek help from a native English speaker for proofreading your text.Importantly, both Reviewers also point out several methodological and statistical flaws that need to be addressed. I especially share Reviewer 1 concern that your conclusions rest on a null effect, which is not so reliable given the small amount of available data. Also, Reviewer 2 worry about the lack of a theoretical definition of "intention" seems extremely relevant to me. Nevertheless, I encourage you to submit a revised version of your manuscript should you feel that these concerns are manageable in a revision.

Please submit your revised manuscript by Sep 28 2024 11:59PM. If you will need more time than this to complete your revisions, please reply to this message or contact the journal office at plosone@plos.org. Please include the following items when submitting your revised manuscript:A rebuttal letter that responds to each point raised by the academic editor and reviewer(s). You should upload this letter as a separate file labeled 'Response to Reviewers'.A marked-up copy of your manuscript that highlights changes made to the original version. You should upload this as a separate file labeled 'Revised Manuscript with Track Changes'.An unmarked version of your revised paper without tracked changes. You should upload this as a separate file labeled 'Manuscript'.

We look forward to receiving your revised manuscript.

Kind regards,

Elisa Scerrati

Academic Editor

PLOS ONE

Journal Requirements:

"The author(s) declare financial support was received for the research, authorship, and/or publication of this article. This work was partially supported by the Japan Society for the Promotion of Science KAKENHI (grant numbers 21K02988, 23K12937) for the equipment, payment to participants, and publication fees."

"The author(s) declare no potential conflicts of interest for the research, authorship, and/or publication of this article."

Reviewers' comments:

Reviewer's Responses to Questions

**Comments to the Author**

1. Is the manuscript technically sound, and do the data support the conclusions?

Reviewer #1: Partly

Reviewer #2: Partly

2. Has the statistical analysis been performed appropriately and rigorously? 

Reviewer #1: No

Reviewer #2: No

3. Have the authors made all data underlying the findings in their manuscript fully available?

Reviewer #1: Yes

Reviewer #2: No

4. Is the manuscript presented in an intelligible fashion and written in standard English?

Reviewer #1: No

Reviewer #2: Yes

5. Review Comments to the Author

Reviewer #1: In this study, the authors explored the impact of virtual reality on the joint Simon effect and, more specifically, the impact of gaze visualization on the amplitude of its effect. To this aim, avatars were used and (own, partner or both) gaze direction was visualized on a mirror placed in front of participants. A significant compatibility effect was only found in the Simon task, and not in a Go/No-Go task, without any further interaction with gaze visualization.

While the objective of the study is interesting, its originality is hampered by a recent publication on the topic (https://doi.org/10.1145/3652920.365293). Moreover, while the investigation of the gaze visualization effect on the joint simon task is new, the methodological flaws present in the proposed article, as well as the little amount of data provided per participant greatly reduce the reliability of the effects. Furthermore, the authors base their conclusion on a null effect of gaze visualization. While null effects are important as well in science, they provide useful information when the methodology followed to obtain such results is impeccable. I find that this is not the case in the proposed study. I will now proceed to detail my objection to the publication of the present article.

Scientific background

The authors specifically investigate the role of gaze visualization on the joint simon effect. The introduction could benefit from citing studies on the role of avatar’s gaze visualization, compared to human gaze, in general, not specifically to the simon effect. Also, the authors should mention a recently publish article on the effect of co-avatar visual representation on the joint simon effect (https://doi.org/10.1145/3652920.365293)

Methods

At page 10 the authors mention that “participants were positioned at the left or right side of the environment”. However, how it was chosen, for each participant to be placed to the left or right of the environment is not specified. Was this factors counterbalanced across participants? Moreover the procedure is not clear. How were participants asked to deliver responses? It is mentioned: “touch a corresponding button”. However, it is not clear to me whether they had to just press a button on the controllers participants held in their hands or on the display?

In experiment 2, procedure paragraph, the authors mention that the order of experimental conditions was “almost” counterbalanced across participants. Why was the order not perfectly counterbalanced?

For both experiments, did participants have a maximum amount of time to respond? Was feedback provided for errors? In the instructions, was speed stressed or accuracy? Not mentioned.

Statistical analyses

Parametric tests were adopted. However, it is not clear whether data was distributed normally. Even for RTs, given the little amount of data per participants, the use of parametric tests can be problematic.

Also, at page 16 the authors mention the use of a 2-way ANOVA for reaction times. Why was a repeated-measures 2 way ANOVA not used, instead, as in the subsequent analysis.

Following the significant main effect of condition on reaction times, the authors further explore the effect of gaze on RTs. They specifically compare the effect of partner-gaze vs. own-gaze visualization. However, this choice is not motivated. It remains unclear why the authors did not include all gaze conditions in this analysis. Specifically, it would have been more interesting to compare partner vs. both gaze visualization. In fact, in real life, people almost never perceive their gaze only.

Results

The results of experiment 1 are of little use. In fact, the authors only used 8 repetitions for each experimental condition. On top of that, the authors mention that outliers and errors were excluded. While mentioning that 91% of total trials were kept, it would be important to include a table with the number of trials, per condition, per participants, that were included in the calculation of the averages used in the analyses. The same thing for experiment 2, in which the number of repetition was augmented to 10, which is more acceptable, but still quite low. Given the little amount of data, it would be appropriate to perform the statistical analyses on medians, instead of means, which are less susceptible to extreme values.

An important difference in the average RTs is present, when comparing results of experiments 1 and 2. The authors do not comment on this. Moreover, the amplitude of the simon effect in experiment 1 is virtually non-existent. The authors do not comment on this.

Discussion

While consistently finding a joint simon effect on both experiment (although the amplitude of this effect greatly differs across the two), the authors did not find a gaze modulation of this effect. The authors proceeded in interpreting this null effect as evidence that i) either visualizing gaze information does not convey partner’s intentions or ii) gaze visualization does not affect the joint simon effect. The authors correctly mention that visualizing gaze direction on a mirror placed in front of participants might have hampered the effects, as well as the fact that a control condition without the partner’s avatar was not included. However, these limitations are not the only possible reasons for this null interaction effect. In fact, the avatars adopted in the study were very basic. It is important for the generation on the joint simon effect that participant feel the presence of the partner and interpret the avatar as conveying humans’ intentions. It is unclear whether participants interpreted the avatar’s choices as coming from a human or not. The latter option could explain the lack of gaze visualization effect.

Moreover, the lateralized placement of the display could greatly affect results. Each participant should be tested twice, in order to control display’s placement on the amplitude of the simon effect. At the least, this factor should be counterbalanced across participants and its effect on the compatibility effect should be tested before any other analyses. It is also not mentioned whether participant were right-handed and which hand was used to deliver responses. The combination of all these factors could account for a simon effect size variability.

As a general remark, English writing is quite bad. Even important words used multiple times in the article such as “square” for the target shape, are consistently misspelled. Revision with a native speaker is suggested for future submissions.

Reviewer #2: The manuscript reports a study in which trough 2 experiments the authors address the JSE in virtual interactions via avatars. I found the topic interesting however I have several concerns about the theoretical frameworks and consequently about the design of the experiments.

I believe that the ms requires major revisions before being considered for publication.

Firstly, starting with the abstract it is not clear what the authors refer at when using the term “intention”. This becomes even more ambiguous in the introduction, as they state on line 62: “… intention of a partner, a belief that a partner perceives stimuli”. However, this is more similar to the definition of Theory or mind and not with that of intentionality or perceiving mental states in a partner. Later on, they review the study by Stenzel et al, in which the authors however manipulated intentionality attribution towards the robot not the perception of TOM (see Marchesi et al 2017 , for a discussion about how an agent can be perceived as intentional even without attributing to it a tom). Such theoretical lack in the definition of intention, is then reflected in the experimental design of experiment 2. Why showing the gaze of the participants themselves or of their partners should have modulated the JSE, if participants were always controlling remotely their avatar, and by task instruction they were obliged to act in a complementary manner, there was no willingness in their actions. On the same issue, later on (see line91), the authors state: ”Given that one’s gaze convey mental states ….. visualizing a partner’s gaze would enhance the intentionality to the perceiver”. This is incorrect, visualizing partner gaze facilitates the decoding of their mental states but does not change whether we perceived them intentional or not-intentional agents. Moreover, the facilitation effect elicited by partner gaze is limited to when the gaze is directed toward a target/goal, otherwise it simply acts as a distracting stimulus that grabs participants attention from the task, or as an alerting one, which I think it may happen in experiment 2 (see below).

Second the introduction lacks in a proper revision of the literature. First, beside the action co-representation account and the referential coding one, there are other accounts. For instance, other accounts started to underline the role of spatial response coding processes (e.g., Guagnano, Rusconi, & Umiltà, 2010; Dolk et al., 2011; Dittrich, Rothe, & Klauer, 2012; Lugli, Iani, Milanese, Sebanz, & Rubichi, 2015; Dittrich, Bossert, Rothe-Wulf, & Klauer, 2017; Dittrich, Pufe, & Klauer, 2017; Michel, Bölte, & Liepelt, 2018; see also see Dolk et al., 2014 and Prinz, 2015 for reviews). In addition, when reviewing literature about the JSE and robots the authors should consider also the work by Ciardo et al 2022, showing that in a JSt ask with a humanoid robot the JSE emerge irrespectively of whether the robot was remotely controlled by the human-partner of by a computer program; and the study by Sahaï et al 2019, showing that the JSE emerges only for humanoid robots but not for servomotor robots.

Third, the results of the two experiments showed that a SE emerged in the go/no-go condition. I see that such effect does not differ from zero, however, as indicated by the lack of significant interaction it does not differ neither from the SE elicited in the Joint conditions of both experiments. The authors completely missed to address such result in their discussion.

MINOR COMMENTS.

On line 110 the authors should better address why the referential coding predicts a lack of JSE when the gaze is displayed.

Line108 the authors should explicitly say that the avatars were always remotely controlled, also in Experiment 1

Where participants only right-handed? If not, please specify how many left-handed participants took part in both Experiment 1 and 2

For replicability it would be better to have a figure or a video displaying the environment used in the syudy

Which was the maximum time allowed for responses? The JSE, as any spatial compatibility effect emerges from the exogenous orienting of attention, and thus it tends to decrease in magnitude for long RTs

Line 169: How long last the blank screen between trials?

Line 177: It would be better to remove from the analysis also the other member of the pair given that the task relies on complementarity

Line 188: RTs are a measure of the time interval between two events (stimulus and response), they do not measure size, so they cannot be “greater than”, but they can be longer or shorter.

When reporting the results it will be better to also report means of each experimental condition instead that simply report a plot. If someone wants to estimate the effect size, it will be impossible to calculate it without means and sd values.

I suggest the authors to put more effort in revising the English writing.

6. PLOS authors have the option to publish the peer review history of their article (what does this mean?). If published, this will include your full peer review and any attached files.

Reviewer #1: No

Reviewer #2: No

---

## [Author Response · Author response to Decision Letter 0]

27 Sep 2024

Plase see the file "Response_to_Reviewers" for specific responses to reviwers' comments.

---

## [Decision Letter · Decision Letter 1]

19 Nov 2024

PONE-D-24-19088R1Effect of Virtual Interactions Through Avatar Agents on the Joint Simon EffectPLOS ONE

Dear Dr. Harada,

Thank you for submitting your manuscript to PLOS ONE. After careful consideration, we feel that it has merit but does not fully meet PLOS ONE’s publication criteria as it currently stands. Therefore, we invite you to submit a revised version of the manuscript that addresses the points raised during the review process.

ACADEMIC I apologize with the authors for the extra time it took me to come back to you with a decision. Reviewer 1 recommended rejection of your manuscript and was not available to consider your revisions. Although I saw many points of their review as valid I believed none was such to preclude publication of your manuscript after a careful revision. Therefore I decided to appoint Reviewer 3 for a further evaluation. Several sharable methodological points were raised by this Reviewer that needs to be carefully addressed. I also warmly recommend you to address the outlier removal and figure reporting points that were highlighted. />==============================

We look forward to receiving your revised manuscript.

Kind regards,

Elisa Scerrati

Academic Editor

PLOS ONE

Reviewers' comments:

Reviewer's Responses to Questions

Comments to the Author

1. If the authors have adequately addressed your comments raised in a previous round of review and you feel that this manuscript is now acceptable for publication, you may indicate that here to bypass the “Comments to the Author” section, enter your conflict of interest statement in the “Confidential to Editor” section, and submit your "Accept" recommendation.

Reviewer #2: All comments have been addressed

Reviewer #3: (No Response)

2. Is the manuscript technically sound, and do the data support the conclusions?

Reviewer #2: Yes

Reviewer #3: (No Response)

3. Has the statistical analysis been performed appropriately and rigorously? 

Reviewer #2: Yes

Reviewer #3: (No Response)

4. Have the authors made all data underlying the findings in their manuscript fully available?

Reviewer #2: Yes

Reviewer #3: (No Response)

5. Is the manuscript presented in an intelligible fashion and written in standard English?

Reviewer #2: Yes

Reviewer #3: (No Response)

6. Review Comments to the Author

Reviewer #2: (No Response)

Reviewer #3: (No Response)

7. PLOS authors have the option to publish the peer review history of their article (what does this mean?). If published, this will include your full peer review and any attached files.

Do you want your identity to be public for this peer review? For information about this choice, including consent withdrawal, please see our Privacy Policy.

Reviewer #2: No

Reviewer #3: No

---

## [Author Response · Author response to Decision Letter 1]

2 Dec 2024

Please see the "Response to Reviewers.doc" file for our replies to the comments by Reviwer #3.

---

## [Decision Letter · Decision Letter 2]

22 Dec 2024

Effect of Virtual Interactions Through Avatar Agents on the Joint Simon Effect

PONE-D-24-19088R2

Dear Dr. Harada,

We’re pleased to inform you that your manuscript has been judged scientifically suitable for publication and will be formally accepted for publication once it meets all outstanding technical requirements.

Kind regards,

Elisa Scerrati

Academic Editor

PLOS ONE

Additional Editor Comments (optional):

Reviewers' comments:

Reviewer's Responses to Questions

**Comments to the Author**

1. If the authors have adequately addressed your comments raised in a previous round of review and you feel that this manuscript is now acceptable for publication, you may indicate that here to bypass the “Comments to the Author” section, enter your conflict of interest statement in the “Confidential to Editor” section, and submit your "Accept" recommendation.

Reviewer #3: All comments have been addressed

2. Is the manuscript technically sound, and do the data support the conclusions?

Reviewer #3: Yes

3. Has the statistical analysis been performed appropriately and rigorously? 

Reviewer #3: (No Response)

4. Have the authors made all data underlying the findings in their manuscript fully available?

Reviewer #3: (No Response)

5. Is the manuscript presented in an intelligible fashion and written in standard English?

Reviewer #3: (No Response)

6. Review Comments to the Author

Reviewer #3: (No Response)

7. PLOS authors have the option to publish the peer review history of their article (what does this mean?). If published, this will include your full peer review and any attached files.

Reviewer #3: No

---

## [Editor Report · Acceptance letter]

28 Dec 2024

PONE-D-24-19088R2 

PLOS ONE

Dear Dr. Harada, 

I'm pleased to inform you that your manuscript has been deemed suitable for publication in PLOS ONE. Congratulations! Your manuscript is now being handed over to our production team.

Kind regards, 

on behalf of

Dr. Elisa Scerrati 

Academic Editor

PLOS ONE